



# CO$_2$ and CH$_4$ fluxes from standing dead trees in a northern conifer forest

Christian Hettwer [1], Kathleen Savage [2], Andrew Ouimette [3], Jay Wason [1], Roel Ruzol [1], Shawn Fraver [1]

[1] School of Forest Resources, University of Maine, Orono, ME 04469, USA
[2] Woodwell Climate Research Center, Falmouth, MA 02540, USA
[3] USDA Forest Service, Northern Research Station, Durham, NH 03825, USA

*Correspondence to*: Christian Hettwer (christian.hettwer@maine.edu)



**Abstract.** Representing 15 – 20% of aboveground biomass in forests, deadwood is an important, yet understudied, component of ecosystem greenhouse gas (GHG) fluxes. In particular, standing dead trees (snags) can serve as conduits for the atmospheric flux of carbon dioxide ($CO_2$) and methane ($CH_4$), with fluxes varying according to environmental conditions. We measured $CO_2$ and $CH_4$ fluxes from six snags along an upland-to-wetland gradient at Howland Research Forest (Maine, USA) with measurements made every two weeks from April to November 2024. Using nonlinear models, we quantified flux responses to environmental predictors including soil moisture, soil temperature, and air temperature. Gas fluxes increased with increasing temperature, yet $CO_2$ flux peaked at moderate soil moisture ($\sim$ 30%), while $CH_4$ peaked at the highest moisture levels. $CH_4$ fluxes were overwhelmingly net positive, suggesting that snags are important pathways for wetland gas emission. $CH_4$ flux was relatively insensitive under low soil moisture and temperature, but increased with rising soil temperature when soil moisture was high, suggesting that methanogenesis depends on anaerobic moisture conditions. Results also suggest that $CO_2$ flux co-varied with $CH_4$ flux from snags, with decreases in $CO_2$ flux associated with increases in $CH_4$ flux. As soil moisture increased, a pronounced shift in gas fluxes (from $CO_2$ to $CH_4$ emission) occurred at $\sim$ 60% soil moisture. These results, which align with those from previous studies establishing anaerobic moisture thresholds and provide new insights into $CO_2$ and $CH_4$ fluxes from snags, present direct measurements of gas exchange from snags along a moisture and temperature gradient.



## 1 Introduction

Deadwood can represent as much as 20% of aboveground biomass in forests, thereby contributing significantly to global carbon dynamics (Komposch et al., 2022; Russell et al., 2015; Woodall et al., 2015). Among deadwood components, standing dead trees, or snags, are particularly important because they remain upright for years, decaying and releasing gases slower than other types of deadwood (Hararuk et al., 2020; Onega and Eickmeier, 1991). This is particularly important in

forests with high snag densities or recent disturbances, where deadwood volumes are elevated (Kipping et al., 2022; Yatskov et al., 2022). Snags also influence carbon flux dynamics by serving as a conduit for atmospheric flux of carbon dioxide ($CO_2$) and methane ($CH_4$), transporting soil-generated gases and also producing within-stem gases (Carmichael et al., 2018). Relatively few studies have addressed greenhouse gas (GHG) fluxes from snags, despite their acknowledged

importance in ecosystem functioning and forest carbon dynamics.

Snags emit $CO_2$ as the result of heterotrophic respiration. As fungi decompose wood, environmental factors, including soil moisture and temperature, influence $CO_2$ flux rates (Mukhortova et al., 2021; Noh et al., 2019). $CO_2$ flux from snags is positively influenced by temperature, as warmer conditions generally enhance decomposition (Renninger et al., 2014).

While moisture is known to have a positive relationship with $CO_2$ flux from deadwood (Olajuyigbe et al., 2012), excessive moisture can limit oxygen diffusion in snag stems, reducing $CO_2$ emissions (Oberle et al., 2018). Snag $CO_2$ emissions (per unit area) are typically smaller in magnitude than those from soils; however, emissions from the two sources are often positively correlated, as both arise from similar biological processes and environmental drivers (Perreault et al., 2021).

Even less well understood is $CH_4$ flux from snags, which can arise through two primary pathways: passive transport of soil-generated $CH_4$ through the stem and *in situ* production within the stem.



In water-saturated soils, methanogenic archaea anaerobically produce $CH_4$, primarily by reducing $CO_2$ (Conrad, 2020). Internal conduits, particularly in trees with compromised xylem structure, can transport $CH_4$ upward through the stem to be diffused to the atmosphere (Keppler et al., 2006;

Pangala et al., 2013). This mechanism is well documented in living trees (Barba et al., 2019; Pangala et al., 2015), and emerging evidence suggests that snags can similarly serve as conduits for soil-emitted $CH_4$ (Carmichael et al. 2018). However, because snags have the potential to host both methanogenic ($CH_4$ emitting) and methanotrophic ($CH_4$ consuming) communities, they may act as net $CH_4$ sources or sinks under certain conditions (Carmichael et al., 2024; Martinez et al.,

2022). In the limited studies available on this topic, the balance between methanogenesis and methanotrophy in snags appears to be strongly influenced by moisture availability and oxygen levels (Terazawa et al., 2021). Thus, in drier, well-aerated upland conditions, methanotrophic activity may dominate, leading to $CH_4$ uptake; in contrast, in saturated lowlands or wetlands, anaerobic conditions support $CH_4$ production, leading to net emission.

Despite their clear contribution to GHG budgets of forests, differences in gas fluxes between living and dead tree stems remain poorly understood. $CH_4$ emissions are generally higher in living than dead stems, depending on stage of decay (Covey et al., 2016; Covey & Megonigal, 2019). $CO_2$ emissions from deadwood are typically higher than those from living stems, but smaller than soil emissions (Warner et al., 2017). Nonetheless, snags remain active sites of biogeochemical

exchange of $CO_2$ and $CH_4$ due to their unique microbial composition. GHG fluxes may also co-vary in ways that reflect underlying environmental controls such as moisture and temperature. Thus, snags may represent a significant, yet often overlooked, component of GHG fluxes in forest ecosystems.



In this technical note, we describe $CO_2$ and $CH_4$ fluxes from repeated measurements on a series of

snags along an upland-to-wetland gradient in a northern temperate conifer forest. Our objectives

were to (1) identify the important drivers of $CO_2$ and $CH_4$ fluxes from snags, (2) illustrate how

these fluxes respond to key environmental variables, and (3) determine if fluxes co-vary along an

upland-to-wetland gradient. In doing so, we clarify the importance of snags in ecosystem carbon

dynamics, their dependence on environmental conditions, and their significance in GHG fluxes.


## 2 Methods

### 2.1 Site Description

This study was conducted at Howland Research Forest of central Maine, USA (45.2041°N

68.7402°W, elevation 60 m above sea level), located in the transition zone between deciduous and

boreal forests in northeastern North America. The climate is damp and cool, with average annual

temperatures of $5.9 \pm 0.8$°C and mean precipitation of $112 \pm 21$ cm/year that is evenly distributed

throughout the year (Daly et al., 2008). Mean daily temperature ranged from -1.2 to 26.6 °C with

an average of 15.2 °C from May to November, 2024, when this study took place. During this same

period, daily precipitation ranged from 0 to 41.3 mm with a mean of 2.5 mm. Peak temperature

occurred on August 2nd and peak precipitation occurred on August 9th.

The mature, multi-aged forest is composed of approximately 90% conifers, primarily red spruce

(*Picea rubens*), which accounts for 51% of the site's basal area. The forest has not been actively

managed since a partial harvest in the 1920s and now displays late-successional features, including

large, old trees (over 200 years), a variety of tree diameters, and diverse stages of coarse woody

debris decomposition (Fien et al., 2019). Soils in the area developed from coarse-loamy granitic





basal till and vary in drainage from well-drained to poorly drained across short distances along

upland-to-wetland transitions (Fernandez et al., 1993). Average soil organic layer depth at the six

snags was 8 cm (Table A1). Snags account for roughly 35% ($440 \pm 20$ g C $m^{-2}$) of deadwood

biomass and 3% of total aboveground biomass at this site (Hollinger et al., 2021).

**2.2 Data Collection**

We randomly selected six standing dead red spruce stems for gas flux sampling (Hettwer et al.,

2025). The snags were within decay classes two and three, based on the five-class system of Sollins

(1982), where one is recently dead wood and five is heavily decomposed. Snags spanned an

upland-to-wetland drainage gradient, where two were in uplands, two in transitional drainage, and

two in an wetlands. Drainage classes (wetland, transitional, upland) were assigned based on soil

moisture data obtained from 100 randomly placed soil moisture sensors, as well as a National

Wetlands Inventory wetland delineation. All analyses were conducted across continuous predictor

variables (e.g., soil moisture, temperature); drainage classes were used solely to guide snag

selection and were not used as grouping factors in the data analyses.

To measure $CO_2$ and $CH_4$ flux from snag surfaces, we affixed custom-fitted PVC collars to snags

with pure silicone at 50 cm stem height. All collars were checked for leaks before each

measurement by exhaling around the collar, with additional silicone applied as necessary. $CO_2$ and

$CH_4$ concentrations were measured by sealing each collar with a custom PVC chamber lid,

connected via 3 meters of 6.4 mm (¼-inch) Bev-A-Line tubing to a LI-7810 analyzer (LI-COR

Biosciences, Lincoln, NE, USA). Ambient air was flushed through the system until gas

concentrations stabilized, after which manual fluxes were recorded at a frequency of 1 Hz for six-

minute intervals (Hutchinson et al., 2000) per observation and processed as described below, along



with spot measurements of air temperature, soil temperature, and volumetric soil moisture (%).

While obtaining flux measurements from each snag, instantaneous air temperature was measured

with radiation shielding (Precision Lollipop Digital Thermometer, Traceable), while soil moisture

and soil temperature were recorded at three locations around the snag at a depth of 15 cm (True

TDR-315N Soil Moisture Sensor, Acclima). While we did not directly measure snag moisture,

previous work has shown that snag volumetric water content closely tracks that of surface soil

(Green et al., 2022). This makes soil moisture, a more widely available and routinely measured

environmental variable, a reasonable proxy for snag moisture in our study. Flux and environmental

data were collected every two weeks from May through November 2024, resulting in a total of 76

measurements. Hemispherical photographs were taken at three points around each snag (two

meters from stem, at bearings 0, 120 and 240°) and processed using Gap Light Analyzer software

(Frazer et al., 1999) to yield canopy openness values. Relative humidity was measured using a

platinum resistance thermometer (EE181-L, Campbell Scientific) from an on-site flux tower, with

data scanned every minute and averaged over 30-minute intervals. For each flux measurement, the

closest corresponding humidity measurement in time was selected.

### 2.3 Data Processing

Fluxes for $CO_2$ ($\mu$mol m$^{-2}$ s$^{-1}$) and $CH_4$ (nmol m$^{-2}$ s$^{-1}$) were calculated from the gas analyzer output

using the ideal gas law (Equation 1), where $PV = nRT$ (P = barometric pressure, atm, V = chamber

volume, L (liters), R (gas constant) = 0.08206 L·atm/mol·K, T = temperature, °C).

$$Flux = \left(\frac{dC}{dt}\right)\left(\frac{V}{A}\right)\frac{P}{(R*(T+273.15))} \tag{1}$$

The rate of change in gas concentration over time (dC/dt) was calculated from 90 to 345 seconds

for each sampling event, using gas concentration versus time data obtained from the gas analyzer.





This timeframe was chosen to reduce the influence of chamber sealing artifacts and initial

concentration surges commonly seen in wetland trees (Yong et al., 2024). Barometric pressure (P)

was paired with the nearest 30-minute measurement recorded by an on-site pressure sensor.

Chamber volumes (V) were determined in the lab for each chamber size by sealing the tree-facing

side with a contour-fitting cover, filling the chamber with quinoa seeds, and measuring the

displaced volume. Surface areas (A) of tree stems within the chambers were also calculated for

each template size. Air temperature (°C) at the time of sampling was converted to Kelvin (K)

before calculating fluxes. The minimum detectable flux was estimated for each measurement

following methods from Christiansen et al. (2015) and Nickerson (2016) with all measured fluxes

falling within detectable limits. Data processing was performed in R (version 4.5.0) (R Core Team,

2024) using RStudio (Posit Team, 2024).


## 2.4 Data Analysis

To investigate the environmental drivers of $CO_2$ and $CH_4$ fluxes from snags, we conducted a series

of statistical and predictive modeling approaches. Random forest analysis, using the *Boruta*

package in R (Kursa and Rudnicki, 2010), allowed us to identify the most influential

environmental predictors associated with each gas flux. These potential predictors included air

temperature, soil temperature, volumetric soil moisture, relative humidity, organic soil pH,

diameter at breast height, and canopy openness. We then used symbolic regression in Python

(PySR) to determine mathematical models that best describe the relationships between top

environmental predictors and gas fluxes (separate models for $CO_2$ and $CH_4$). This algorithm

generated interpretable nonlinear models by automatically developing numerous candidate models

and ranking them based on AICc scores (Cranmer, 2023).

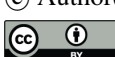



To evaluate the strength of nonlinear interaction effects between predictor variables, we computed the second-order mixed partial derivatives ($\partial^2 y / \partial x_1 \partial x_2$) of the fitted nonlinear response surfaces. Doing so allowed us to assess how the effect of one predictor (e.g., soil moisture) on the response

(e.g., $CO_2$ or $CH_4$ flux) changed depending on the level of another predictor (e.g., soil or air temperature). By calculating these interaction terms across a grid of observed data ranges, we identified regions where interactions were strongest, which may indicate synergistic or antagonistic environmental effects on gas fluxes. Given the response curve, we determined the optimal soil moisture ($x_1$) for maximizing $CO_2$ flux ($y$) by setting the derivative of $\ln(y)$ with

respect to $x_1$ to 0 (Equation 2).

$$\frac{d}{dx_1}\ln(y) = ab - 2a\frac{x_1}{x_2} = 0 \;\rightarrow\; x_1 = \frac{bx_2}{2} \tag{2}$$

To determine if the relationship between $CO_2$ and $CH_4$ fluxes varied across the soil moisture gradient, we fit linear models with $CH_4$ flux as a response to the interaction between soil moisture and $CO_2$ flux. For $CO_2$ and $CH_4$ fluxes in this two-gas model, we used our top-performing

nonlinear models and held temperature constant at low (25th percentile) and high (75th percentile) values. We then used a multivariate analysis of variance (MANOVA) to test whether soil moisture jointly influenced $CO_2$ and $CH_4$ fluxes, treating both gases as simultaneous response variables, at low and high temperatures. Lastly, we used Pearson tests on raw data to determine the correlation between $CO_2$ and $CH_4$ fluxes.


### 3 Results

Sampling period snag $CO_2$ fluxes ranged from 0.20 to 9.75 μmol m$^{-2}$ s$^{-1}$ (mean = 2.26 ± 2.34) and $CH_4$ fluxes ranged from -0.21 to 2.46 nmol m$^{-2}$ s$^{-1}$ (mean = 0.25 ± 0.44). $CO_2$ emissions were highest from late-June to late-July and $CH_4$ emissions were highest from mid-July to mid-August





(Fig. 1). Minor $CH_4$ uptake was observed only three times out of 72 measurements throughout the

sampling period, all of which occurred on upland snags. Mean $CO_2$ and $CH_4$ fluxes approached

zero between October (day of year 275) and November.

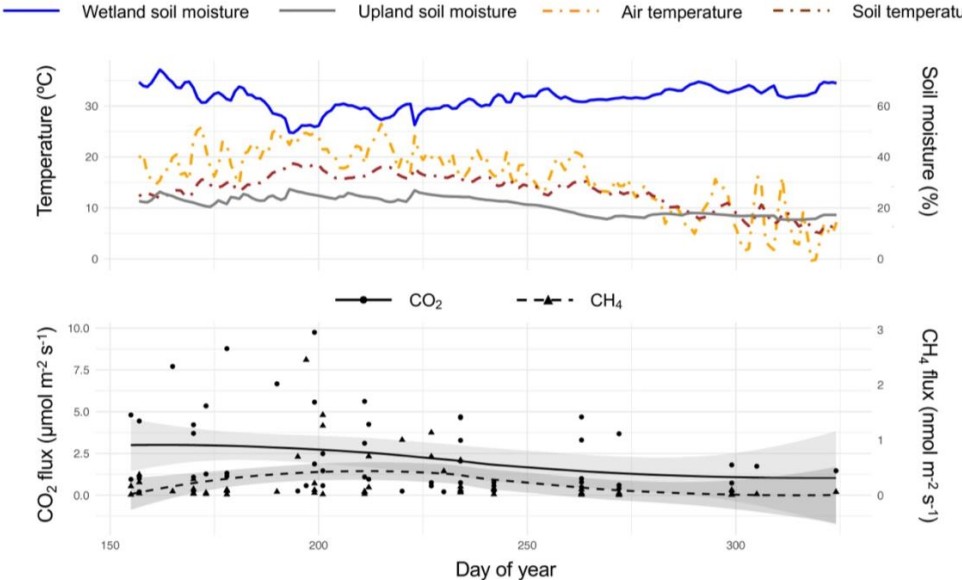

**Figure 1: Top pane shows environmental conditions (temperature and soil moisture) throughout the sampling period, obtained from on-site eddy-covariance tower. Bottom**
**pane shows observed $CO_2$ (primary y-axis) and $CH_4$ (secondary y-axis) fluxes vs. day of year. The shaded regions around each spline represent the 95% confidence interval, calculated using standard errors from the smoothing model.**

Based on random forest analysis, soil moisture, air temperature, and canopy openness were

deemed important for $CO_2$ flux, and soil moisture, soil temperature, and canopy openness for $CH_4$

flux. Canopy openness was strongly correlated with soil moisture (r = 0.74) and was therefore

omitted from subsequent analyses. Relative humidity and stem diameter were classified as

unimportant and were also omitted. Results from our nonlinear model fitting suggested that $CO_2$

flux followed an exponential increase or decrease depending on the relative values of soil moisture

and air temperature (Table 1, Fig. 2). Importantly, this model has the potential for negative, or

antagonistic, interaction between predictors. $CH_4$ flux responded to soil moisture and soil




**Table 1:** Top-performing nonlinear models for $CO_2$ flux ($\mu$mol m$^{-2}$ s$^{-1}$) and $CH_4$ flux (nmol m$^{-2}$ s$^{-1}$) from snags with the functional forms of each model, the estimated parameters, and associated coefficients of determination ($R^2$).

| Response ($y$) | Predictors ($x_1$, $x_2$) | Model form | $a$ | $b$ | $R^2$ |
|---|---|---|---|---|---|
| $CO_2$ flux | Soil moisture, Air temperature | $y = e^{a \cdot x_1 \cdot (b - \frac{x_1}{x_2})}$ | 0.0657 | 2.119 | 0.62 |
| $CH_4$ flux | Soil moisture, Soil temperature | $y = \dfrac{x_2 + a}{b - x_1}$ | -7.0015 | 89.999 | 0.90 |

temperature as a rational function where flux changed slowly until soil moisture approached a

critical value, above which it increased dramatically, although modulated by soil temperature

(Table 1, Fig. 2).

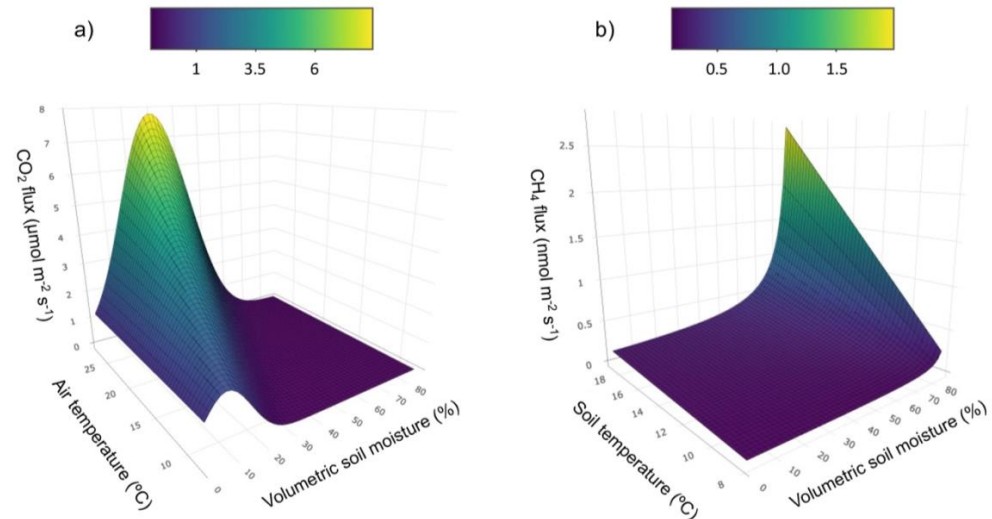

**Figure 2: Modeled relationships between (a) $CO_2$ and (b) $CH_4$ snag fluxes and top environmental variables, highlighting the important interactions between predictors for each**
**gas. Color scales correspond to $CO_2$ ($\mu$mol m$^{-2}$ s$^{-1}$) and $CH_4$ (nmol m$^{-2}$ s$^{-1}$) units.**

Modeling based on partial derivatives highlighted distinct response surfaces for the two gases

while revealing interactive effects between the predictors for each (Fig. 2). The highest $CO_2$ fluxes

occurred at intermediate soil moisture levels ($25 - 35\%$ volumetric soil moisture) and elevated

temperatures ($> 20$ °C). The strongest positive interaction for predicting $CO_2$ flux ($\partial^2 y / \partial x_1 \partial x_2 =$

$0.05 \frac{\mu\text{mol m}^{-2}\,\text{s}^{-1}}{\%VWC \cdot °C}$) occurred at 27% soil moisture and 28 °C air temperature (maximum observed





value). The strongest negative interaction occurred at 53% soil moisture and 28 °C air temperature. Assuming maximum observed air temperature, $CO_2$ flux reached its maximum at 30% volumetric soil moisture. The nonlinear models also highlight the relationship between $CO_2$ and $CH_4$ fluxes along the upland-to-wetland moisture gradient (Fig. 3). $CO_2$ flux peaked at approximately 25%

soil moisture, then declined exponentially until becoming negligible at approximately 60% soil moisture, where $CH_4$ fluxes exhibited a sharp increase.

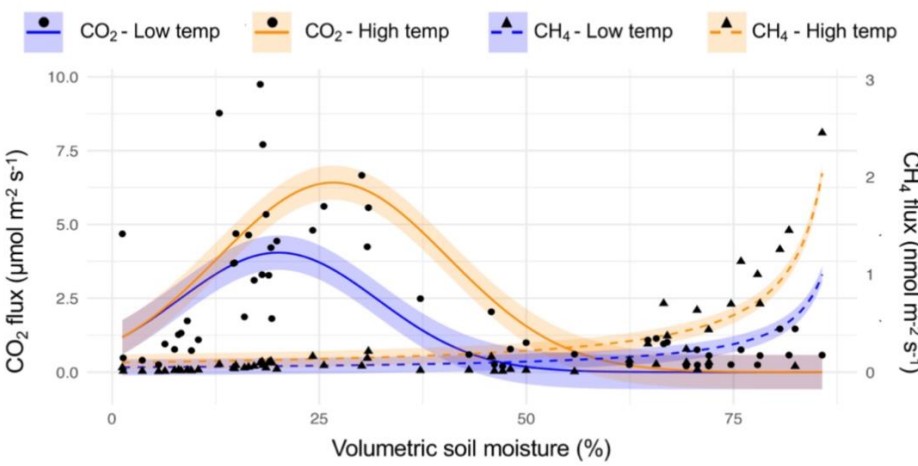

**Figure 3: Observed snag $CO_2$ and $CH_4$ fluxes along a soil-moisture gradient, with curve fits corresponding to top-performing nonlinear models. Temperature (air temperature for $CO_2$, soil temperature for $CH_4$) was held constant at low (25th percentile) and high (75th percentile)**

**values for predicting curves. Shaded regions around each curve represent the 95% confidence intervals.**

$CH_4$ fluxes increased sharply with higher soil moisture levels (~ 60%) and exhibited negligible response to increases in soil temperature, except when soil moisture was high. The strongest interaction effect between predictors was observed at 86% soil moisture (maximum observed

value) and 17 °C soil temperature, while the weakest interaction occurred at 1% soil moisture (minimum observed value) and 14 °C soil temperature.





The two-gas model revealed a significant negative interaction between $CO_2$ flux and soil moisture in predicting $CH_4$ flux ($p < 0.001$), regardless of temperature percentile, indicating that the relationship between the two gases varied along the soil moisture gradient. Specifically, $CH_4$ flux

increased when $CO_2$ flux decreased under increasing moisture conditions. Furthermore, the MANOVA results show that $CO_2$ and $CH_4$ fluxes were jointly influenced by soil moisture ($p < 0.001$). $CH_4$ and $CO_2$ fluxes were negatively correlated ($r = -0.53$), suggesting they respond differently along the soil moisture gradient.

**4 Discussion**

Our results highlight the strong influence that soil moisture and temperature exert on gas fluxes from snags, demonstrating these responses along an upland-to-wetland moisture gradient. $CO_2$ fluxes peaked at intermediate soil moisture ($\sim 30\%$) and high temperatures, showing a significant interaction between soil moisture and air temperature. $CH_4$ fluxes rose sharply at high soil moisture

($\sim 60\%$) and were primarily driven by soil moisture, with a strong interaction between soil moisture and soil temperature.

$CO_2$ flux increased as air temperature increased, in agreement with previous studies (Boddy, 1983; Noh et al., 2019), but it also depended on soil moisture. As soil moisture deviated in either direction from that of maximum $CO_2$ flux ($\sim 30\%$), flux decreased towards zero. Therefore, our results

suggest that fungal activity in snags, and in turn $CO_2$ emissions, may be inhibited at both low and high moisture conditions. The finding of reduced deadwood $CO_2$ flux under low moisture conditions has been reported in previous studies (Boddy, 1983; Hicks et al., 2003). The finding of reduced $CO_2$ flux at high moisture conditions corroborates several deadwood studies (Meyer and



Brischke, 2015; Progar et al., 2000) but contrasts with others (Forrester et al., 2012; Gough et al., 2007).


$CH_4$ flux increased sharply with rising soil moisture when soil temperature was high, indicating a synergistic effect between the two variables. Similarly, $CH_4$ flux became more responsive to increases in soil temperature when soil moisture was high, suggesting that optimal conditions for $CH_4$ emission occurred when both predictors were elevated. $CH_4$ flux rates, as well as interaction effects of predictor variables, were negligible in drier, cooler conditions. The insensitivity of $CH_4$ flux to increasing soil moisture, even at high temperatures, aligns with findings of a volumetric soil moisture threshold of ~ 60% beyond which $CH_4$ flux rates from soils increase dramatically (von Fischer and Hedin, 2007). While a positive relationship between $CH_4$ flux and moisture has been observed from soils and other deadwood forms (Covey & Megonigal, 2019; Kipping et al., 2022), the dynamics of snag $CH_4$ flux and its dependence on moisture remain poorly understood, perhaps due to the challenge of directly measuring internal snag moisture (Green et al., 2022).



Our results point to a complex, moisture-sensitive relationship between $CO_2$ and $CH_4$ fluxes in the studied ecosystem. The significant interaction between $CO_2$ flux and soil moisture in predicting $CH_4$ flux suggests that the behavior of one gas flux may be linked to the other, especially under varying moisture conditions. As soil moisture increased, $CO_2$ flux declined while $CH_4$ flux rose, indicating a possible transition from aerobic to anaerobic microbial processes. The negative correlation between $CO_2$ and $CH_4$ fluxes, combined with MANOVA results showing soil moisture as a joint flux driver, reinforces the assumption that the coupling of these gas fluxes is moisture dependent. Furthermore, the shift in positive $CO_2$ to $CH_4$ fluxes between 60 – 70% moisture aligns with reported values for volumetric water content at which conditions shift from aerobic to anaerobic (Długosz et al., 2024; Fairbairn et al., 2023; Schlüter et al., 2025). Overall, these findings







underscore the importance of soil moisture in regulating greenhouse gas dynamics in forested systems and the value of using a two-gas framework to understand GHG fluxes as they pertain to radiative forcing and climate change.

In summary, we found that moisture and temperature interact in complex ways to govern both $CO_2$ and $CH_4$ fluxes from snags. Both gas fluxes increased with increasing temperature, yet $CO_2$ flux peaked at moderate moisture levels, while $CH_4$ peaked at the highest moisture recorded. $CH_4$ fluxes were overwhelmingly net positive (i.e., sources to the atmosphere), suggesting that snags represent an important pathway for wetland gas production; however, they may serve as $CH_4$ sinks

under certain conditions. Our results derive from a small number of snags, yet they provide compelling evidence for the importance of snag fluxes in the forest carbon cycle, particularly considering projected increases in regional temperature and precipitation (Fernandez et al., 2020). Our results identify further knowledge gaps regarding the influence of snag species (via wood anatomy), stem height, and stages of snag decay. Our results also point to the need for methods of

measuring internal snag moisture, rather than relying on soil moisture as a proxy. Finally, phylogenetic sequencing of the bacterial and fungal communities residing in snags would clarify the mechanisms driving the variations in $CO_2$ and $CH_4$ fluxes from snags.

**Conclusions**

These results demonstrate that standing dead trees (snags) in northern conifer forests emit both $CO_2$ and $CH_4$, with flux dynamics strongly regulated by environmental gradients, particularly soil moisture. $CO_2$ flux peaked at intermediate soil moisture (~ 30%) and warm air temperatures, following a rational response shaped by interactive effects between soil moisture and air temperature. In contrast, $CH_4$ flux remained low until soil moisture exceeded a threshold (~ 60%),



after which emissions increased sharply, modulated by soil temperature. The observed negative

correlation between $CO_2$ and $CH_4$ fluxes, along with evidence of antagonistic interactions between

$CO_2$ flux and soil moisture in predicting $CH_4$ flux, suggests that the two gases respond divergently

along the upland-to-wetland gradient. Overall, these findings highlight the complexity of

greenhouse gas emissions from snags and underscore the importance of considering interactive

environmental drivers when modeling their contributions to forest carbon dynamics.

**Appendix A**

**Table A1:** Organic layer depth (mean of three points) for each snag and mean soil moisture and
snag fluxes across the sampling period. Also reported are the standard deviations for soil moisture,
$CH_4$ flux, and $CO_2$ flux.

| Snag | Mean soil organic layer depth (cm) | Mean soil moisture (%) | Mean $CH_4$ flux (nmol m$^{-2}$ s$^{-1}$) | Mean $CO_2$ flux (µmol m$^{-2}$ s$^{-1}$) |
|---|---|---|---|---|
| 1 | 7.62 | 67.9 ± 12.9 | 0.628 ± 0.708 | 0.536 ± 0.492 |
| 2 | 10.16 | 64.4 ± 12.9 | 0.493 ± 0.444 | 0.875 ± 0.395 |
| 3 | 8.23 | 19.2 ± 9.5 | 0.085 ± 0.028 | 5.070 ± 3.040 |
| 4 | 9.53 | 23.1 ± 5.7 | 0.133 ± 0.043 | 3.890 ± 1.220 |
| 5 | 5.56 | 7.46 ± 4.2 | 0.017 ± 0.011 | 0.933 ± 0.519 |
| 6 | 6.99 | 15.6 ± 6.9 | 0.045 ± 0.017 | 4.590 ± 1.580 |

**Data availability**
The datasets generated and analyzed during the current study have been made available in the
Environmental Data Initiative (EDI) repository. Refer to the citation for Hettwer et al., 2025 in the
references.

**Competing interests**
The authors declare that they have no conflicts of interest.

**Acknowledgements**
We would like to thank Ivan Fernandez, Jonathan Gewirtzman, and Rachel Poppe for their help
with this manuscript. This research was supported in part by the U.S. National Science Foundation
(DEB Award #2208658, 2208655), the U.S. Department of Energy's Office of Science (AmeriFlux
core site), the U.S. Forest Service, Northern Research Station (JVA #25-JV-11242306-009), and
the Maine Agricultural and Forest Experiment Station (ME042612, ME042121).



**Author contributions**

All authors contributed to the study conception and design. Material preparation, data collection and analysis were performed by Christian Hettwer. The first draft of the manuscript was written by Christian Hettwer and all authors commented on previous versions of the manuscript. All authors read and approved the final manuscript.



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
