# Peer review of "CO2 and CH4 fluxes from standing dead trees in a northern conifer forest"

_EGUsphere, 2025_

## Author Response (AR1)

Message to the handling editor

We thank you and the referees for the time and effort devoted to reviewing our manuscript. Your suggestions have substantially improved the clarity and impact of the work. In this revised version, we have thoroughly addressed all referee comments and incorporated gas flux measurements from additional substrates at our research site, collected during the same growing season under consistent protocols. Together with a comparison to fluxes reported in the literature, these additions further strengthen the paper and reinforce the importance of snags in forest GHG fluxes.

We would also like to note that we have included the 2D heatmap in Figure 3, as recommended by Referee #2. While we are happy to retain it if preferred, we believe it does not enhance interpretation beyond what is already conveyed in the 3D surface plot. The 3D plot more effectively visualizes both the spatial "hotspots" of emissions and the structure of the nonlinear models, whereas the 2D plot reflects only the former.

Responses to referees' comments on the submitted manuscript, "CO2 and CH4 fluxes from standing dead trees in a northern conifer forest"

Referee #1

General Comments:

This study investigates spatiotemporal patterns of CO2 and CH4 fluxes from standing dead wood in a North American forest. It provides insights into the relationships of these fluxes with local temperature and soil moisture conditions.

I do not have any technical issues with this paper, but it is not a very compelling narrative. While looking at standing snags instead of fallen deadwood is interesting, the analyses are quite basic and reveal well-established relationships between temperature and moisture (which is just soil moisture in this study) and CO2 and CH4 fluxes from decaying biomass. They do not place their findings in a broader ecological context (e.g., how significant are snag fluxes to overall ecosystem fluxes?), and there is not a lot of analysis of the snag wood itself, which could reveal more interesting and novel mechanisms of GHG production. Similarly, there is likely substantial heterogeneity of fluxes within any given snag, but this is not investigated. It's as if there are logical next steps in this research that were not taken, and the impact of the work is diminished as a result.

With all that said, I think the writing style and clarity of the manuscript are good. The figures are well constructed. The methods are reproducible.

*RESPONSE: We thank the reviewer for their thoughtful and constructive feedback. We appreciate that they found the manuscript well written, clearly presented, and methodologically*

*sound. We also value their insights on how the narrative could be strengthened by better contextualizing our findings and considering additional aspects of heterogeneity and mechanistic processes. Below, we address these points in detail.*

**1. Broader ecological context (ecosystem-scale significance of snag fluxes)**
*We agree that placing snag fluxes within the broader framework of ecosystem carbon dynamics enhances the manuscript's impact. In the revised version, we will include a section in the Discussion that compares our observed $CO_2$ and $CH_4$ fluxes to those from soil, other coarse woody debris (i.e. logs and stumps), and living tree stems at Howland Forest. We will do this using data collected during the same growing season (2024). This comparison underscores the potential contribution of standing dead wood to total ecosystem greenhouse gas budgets, particularly in systems where snag volume is high due to disturbance or mortality events.*

**2. Limited analysis of snag wood properties and internal heterogeneity**
*We appreciate the reviewer's suggestion to explore within-snag variation and wood characteristics, as these factors likely influence gas production and diffusion. Although detailed internal wood analyses were beyond the scope of the present study, we have expanded the Discussion to acknowledge this as a key direction for future work. Specifically, we now discuss how wood density, moisture gradients, fungal colonization, and internal anoxia could generate spatial variability in fluxes along the snag stem (e.g., vertical gradients or bark vs. sapwood differences).*

**3. Perceived simplicity of the analyses**
*We understand the reviewer's observation that temperature and moisture are well-established drivers of wood decomposition and greenhouse gas fluxes. Our intention was not to redefine these relationships, but to quantify how they operate specifically in standing dead wood—an understudied substrate type that experiences distinct microclimatic and hydrological conditions compared to downed logs or soil. Our results reveal that the strength and direction of these relationships differ between $CH_4$ and $CO_2$, highlighting a two-gas framework for assessing GHG fluxes.*

**4. Future research directions**
*We agree that the next logical steps involve (i) incorporating wood physical and microbial properties to understand mechanistic controls, and (ii) upscaling snag fluxes within ecosystem carbon budgets. We now close the Discussion by outlining these directions, emphasizing that this study represents an essential first step in quantifying and characterizing snag GHG dynamics—a component often neglected in terrestrial carbon models.*

Specific Comments:

*Site Description: Including a plot of the study site and zones of different soil drainage classes in the main text would provide helpful context for the readers.*

*RESPONSE: We agree that a plot of the study site and zones of different soil drainage classes in the main text would provide helpful context, and we will include this in the revised version of the manuscript.*

Referee #2

General comments:

This study conducted a six-month time series of in-situ measurements of $CO_2$ and $CH_4$ fluxes in a Maine forest, along with measurements of key environmental predictors. The results indicated that soil moisture is the primary driver of both gases' fluxes, while temperature also plays a role.

The study highlights the importance of $CO_2$ and $CH_4$ fluxes from snags to the atmosphere. However, even though information on the biomass percentage of the snags was provided at the beginning, no quantitative comparison on the actual $CO_2$ or $CH_4$ fluxes were made after they got the fluxes data. I would encourage the authors to include reference $CO_2$ and $CH_4$ fluxes, especially of different forest ecosystems, to allow for a more direct and robust comparison.

*RESPONSE: We appreciate the reviewer's helpful feedback on our manuscript. As stated in our response to referee #1, we will include a comparison of CO2 and CH4 fluxes from standing dead trees to those from soils, other coarse woody debris, and living trees collected during the same growing season at our research site. This, paired with comparisons to other publications' flux results from snags (spanning different ecosystem types) in our Discussion section, should emphasize the importance of our findings and better contextualize the results.*

Specific Comments:

Line 199-201, I suggest adding some explanation for why $CO_2$ flux is influenced by air temperature while $CH_4$ flux is influenced by soil temperature. Currently, the manuscript only states that this comes from the random forest analysis, which lacks the science behind it.

*RESPONSE: We agree and will address this mechanistically by stating methanogenesis occurs largely in wet soils, while $CO_2$ production occurs in the wood itself (in Discussion section).*

Lines 113–135: For readers who have not conducted this type of research, a schematic illustrating how the flux measurements were conducted would be very helpful.

*RESPONSE: We agree and will include an image of the chamber design and IRGA setup.*

Figure 2: A 2D heatmap may be more straightforward than a 3D contour plot to show how volumetric soil moisture and temperature influence $CO_2$ and $CH_4$ fluxes.

*RESPONSE: We agree that a 2D heatmap is an effective way of showing moisture and temperature ranges where $CO_2$ and $CH_4$ fluxes are high or low, but we contend that the 3D contour plot better represents the response trends as well as the results of our symbolic regression modeling. We're open to including the 2D heatmaps, in addition to our current 3D contour plots, to reinforce the interpretation of our gas fluxes.*

Lines 222–226: The description of maximum $CO_2$ flux is a bit confusing. Line 222 states, "Assuming maximum observed air temperature, $CO_2$ flux reached its maximum at 30% volumetric soil moisture," whereas line 224 says, "$CO_2$ flux peaked at approximately 25% soil moisture, then declined exponentially until becoming negligible at approximately 60% soil moisture." The manuscript should clarify which soil moisture corresponds to the maximum flux.

*RESPONSE: We agree that we made an error here and will correct in the revised manuscript to ensure consistency.*

Lines 237–238: "The two-gas model revealed a significant negative interaction between $CO_2$ flux and soil moisture in predicting $CH_4$ flux ($p < 0.001$)" is somewhat misleading, as it sounds like soil moisture negatively affects $CH_4$ flux, while the results actually show a positive effect.

*RESPONSE: We agree and will revise this sentence as such: "The two-gas model revealed a significant negative interaction between $CO_2$ flux and soil moisture in predicting $CH_4$ flux ($p < 0.001$), indicating that the positive effect of soil moisture on $CH_4$ flux weakened as $CO_2$ flux increased."*

Lines 305–307: similar to the above comment, the phrase "antagonistic interactions between $CO_2$ flux and soil moisture in predicting $CH_4$ flux" is confusing and seems contrary to the results. It may be clearer to rephrase this to reflect the actual relationship observed between the two gases along the upland-to-wetland gradient.

*RESPONSE: We agree that this phrasing is a bit confusing and have rephrased the sentence as such: "The observed negative correlation between $CO_2$ and $CH_4$ fluxes, along with evidence of an interaction between $CO_2$ flux and soil moisture in predicting $CH_4$ flux, suggests that the two gases respond divergently along the upland-to-wetland gradient."*

---

## Author Response (AR2)

Message to the handling editor

We thank you and the referee for your continued effort to review our manuscript. We believe your feedback has improved the manuscript, making it a better contribution to your journal and to the scientific community at large.

We have accepted all of the referee's comments, which should be reflected in the revised manuscript.

Responses to referees' comments on the submitted manuscript, "CO2 and CH4 fluxes from standing dead trees in a northern conifer forest"

Referee #1

Specific Comments:

L42: Adding a quick reference to the sorts of disturbances that could lead to a snag-rich forest (e.g., disease/pests, drought) would help reinforce the importance of this work. Such mortality events will likely become more common in the future.

*Response: We agree and have added three references and lines in the introduction, discussion, and conclusions related to increased forest disturbance and hence snag abundance. This will help to clarify the scope and importance of our results.*

Conclusions: To me, the conclusions largely show that CO2 and CH4 fluxes from snags behave similarly to those from soils. This makes sense, as snags are carbon-rich substrates just like soils (even more carbon rich). But, the authors do not mention these similarities here, which was confusing to me. I also feel that the authors could note potential knowledge gaps and avenues for future research here as well.

Response: We agree and have modified the Conclusions section to include a comment on similarity to soils, as well as knowledge gaps and avenues for future research.